# Formation of Oxides and Sulfides during the Welding Process of S700MC Steel by Using New Electrodes Wires

**DOI:** 10.3390/ma17122974

**Published:** 2024-06-18

**Authors:** Bożena Szczucka-Lasota, Tomasz Węgrzyn, Adam Jurek

**Affiliations:** 1Faculty of Transport and Aviation Engineering, Silesian University of Technology, Krasinskiego 8, 40-019 Katowice, Poland; 2Novar Sp. z o. o., Towarowa 2/8, 44-100 Gliwice, Poland; adam.jurek@novar.pl

**Keywords:** high-strength steel, HSS, MnS formation mechanism, welding, mechanical resistance, microstructure, automotive industry welded construction

## Abstract

To receive a high-quality welding structure of high-strength S700MC steel for applications in the automotive industry, newly developed electrode wires with increased silicon and manganese content were used. The strength and structural tests of the obtained joints were performed. In the weld, we identified the beneficial oxides strengthening the joint structure and unfavorable MnS inclusions. The non-metallic inclusions were formed inside the weld. Their arrangement, morphology, and chemical composition is described. A view on the high-temperature mechanisms of the formations included during the welding process with new electrode wires is presented. It was found that the dominant mechanism of the inclusion formation and the temperature of the welding process impact the content and varied morphology of inclusions, thus determining the exploitation time of the welded joints obtained. The obtained MAG joints made S700MC steel, due to the formation mainly of oxide inclusions and a relatively small amount of MnS phase, were characterized by a high value of yield and tensile strength, which makes them a promising solution for the automotive industry, especially against the background of connections from the discussed steel grade presented in the literature.

## 1. Introduction

So far, the most critical construction material used in producing car bodies and other elements in means of transport is steel. This material is available, and it is characterized by sufficiently high tensile and compressive strength. Its technological processing is easy. Due to the high density of metal alloys compared to some composites or aluminum alloys, the weight of steel structures is high. Weight reduction is ensured, among other things, using thin-walled profiles and sheets [1,2]. The development of steel materials, in this area, aims to obtain their high-strength characteristics, ensuring appropriate mechanical properties while obtaining thinner profiles of the structure walls. Currently, the car bodies sheets have a thickness of 0.6 to 2.5 mm. The selection of sheet thickness depends on the grade and properties of the steel, the location of the element during operation, and the function of the structure [3,4]. Currently, in addition to unalloyed steels, the following steel grades are used in the construction of means of transport [5,6,7]:Ferritic–martensitic steels (DP—Dual Phase);TRIP (TRIP—Transformation-Induced Plasticity) steels;Austenitic steels;Duplex steel;Martensitic steel.

In responsible zones, i.e., those that largely determine the passive safety of vehicles, the latest construction solutions are introduced from materials of high functional quality, e.g., high or ultra-high strength, including the following [8,9,10]:High-strength steel (HSS);Advanced high-alloy steel (AHSS) (Figure 1).

Both material groups have a dominant martensitic microstructure, which creates significant welding problems. For welded structures of motor vehicle bodies to be characterized by high performance properties and ensure operational safety, joints must meet a few quality requirements. The authors of various scientific studies are looking for the possibility of proper welding of high-strength steel to obtain the following [12,13,14,15]:Acceptable properties of the joint;Uniform hardness in the entire welded joint;High strength of the connection without welding defects. It should be emphasized that the martensitic structure of the latest high-strength steels used in the automotive industry is hard and brittle, which promotes the formation of cracks in the joint and significantly hinders the selection of welding process parameters.

The literature on the subject presents various solutions to produce a safe welded structure from the discussed steel grades, which are intended for applications in the automotive industry [16,17,18,19]. The first solutions focused on obtaining joints without welding defects. Most often, using a temperature regime, appropriately selected for a given type of steel, and maintaining other welding parameters ensured correct joints but with strength characteristics much lower than those of the base material. This solution limited using high-strength steels in some welded structures, e.g., for automotive applications. Therefore, new investigations have been focused on the obtaining joints with mechanical properties similar to the base material. The solutions consist of modifying the existing welding processes and their parameters. For example, attempts were made to use electrode wires with an increased content of titanium and niobium to obtain non-metallic inclusions to strengthen the weld.

The solution enabled the fabrication of structures characterized by non-metallic inclusions such as oxides, nitrides, titanium carbides or niobium [20,21]. Inclusions of various shapes and sizes were observed in the weld. The inclusions on the one hand improved the tensile strength but on the other hand reduced the fatigue strength of the joint. This view is consistent with information in the literature [22,23,24]. The performance of steel has an important connection with MnS inclusions. Different forms of MnS will produce different effects. Because chain inclusions and long inclusions can cause defects in the steel, it is important to try to control the MnS inclusions into fusiform or spherical shapes. When the total oxygen content in steel is low, it is easy to form pure MnS inclusions with good plasticity. The MnS inclusions can be significantly extended; in the process, they can affect the mechanical properties. When the total oxygen content in steel increases, the number of inclusions containing Al_2_O_3_ increases greatly. And the Al_2_O_3_ core MnS heterogeneous nucleation complex inclusions increased in proportion, as shown in the literature presented below. The average size of this type of inclusion is smaller and harder than the MnS inclusions alone, and the possibility of deformation in the rolling process is lower.

Different authors have tried to use various shielding gases, including argon gas mixtures with nitrogen or oxygen additives, to increase the fatigue strength of the resulting joint. However, the solutions did not provide comparable hardness in all zones of the joint [25,26,27].

Therefore, in the article, it was decided to evaluate the newly developed electrode wires with a significantly increased silicon content to about 1%, because the weld metal of such a wire has an immediate strength of about 700 MPa, which is comparable to the value of this mechanical parameter of S700MC steel. The article discusses very important issues related to the influence of elements (silicon, manganese, oxygen) on the strength properties of welded joints Based on the [22,23,24] literature analysis, it was concluded that the increased silicon content into the weld metal will have a positive effect on the strength of welded joints, primarily through the formation of non-metallic silicate inclusions and the accompanying strengthening effect. Silicon has a high affinity for oxygen and is an effective deoxidizer. The resulting SiO_2_ and MnO inclusions strengthen the weld and facilitate the nucleation of MnS sulfides in contact with them.

It was decided to increase the manganese content and reduce the sulfur content in the weld metal, too. This solution enables the formation of slight, favorable inclusions, such as oxides and oxide spinels from the MnO-Al_2_O_3_ system, as well as the reduction in the number and growth of unfavorable MnS inclusions.

In the manuscript, it was checked whether the newly developed electrode wires enable correct welded joints to be made and whether their chemical composition will affect the quality and strength of the obtained joints. Additionally, the mechanism of inclusion formation during welding processes using newly developed electrode wires was determined. These mechanisms influence the strengthening of the weld.

## 2. Materials and Methods

For the MAG welding of S700MC steel (Table 1), two newly developed laboratory wires were used, which were made to contain increased silicon and manganese content:0.07% C, 1.6% Mn, 0.9% Si (X16 wire);0.07% C, 1.4% Mn, 0.9% Si 9 (X14 wire).

The diameter of both wires was 1 mm.

Welding was carried out using an automatic welding machine. The sheet metal used in the process was checked (Table 1) before the welding process.

The data in Table 1 show that the high-strength steel used in the tests is characterized by a content of niobium, titanium at a comparable level of about 0.1% and a silicon content of about 0.9%. Our test results (Table 1) show that the high silicon content is comparable to that of laboratory-developed electrode wires.

The joints of the S700MC steel sheets were made in an Ar-18% CO_2_ gas mixture and preheated to a temperature of 115 °C. In HSS steels, preheating dries the joint and eliminates the hydrogen content in the weld. This temperature should be above 100 °C. Additionally, preheating slows down the cooling of the joint, which translates into the elimination of welding defects.

Then, 2 mm thick welding samples were prepared without chamfering. The weld was formed as a single-legged weld. At the beginning of the welding process, the following current and voltage values were proposed:Welding current: 114 A;Arc voltage: 22 V.

Other important welding parameters assumed the following values:Welding speed: 310 mm/min;Shielding gas flow: 14 dm^3^/min (customarily 14 L/min).

The most important element of the study was to test MAG welded joints obtained by two types newly developed electrode wires.

After the welding process with different parameters, non-destructive testing (NDT) and destructive testing (DT) were carried out. In the case of NDT tests, the following were carried out:VT—visual examination in accordance with the requirements of PN-EN ISO 17638:2017-01 [28];MT—magnetic particle testing, fulfilling the guidelines: PN-EN ISO 5817:2023-08 [29], using a magnetic flaw detector device type REM-230 (SOLYSOL, Toledo, Spain).

Then, the samples for macroscopic tests (criteria according to EN ISO 5817 standard), hardness measurements, and for selected mechanical testing were prepared. The samples were cut out of welded sheets according to the scheme shown in Figure 2. The article is limited to presenting the results of mechanical tests, such as the following:Tensile strength test, carried out by EN ISO 527-1:2019 [30] using an Instron 8874 machine (Instron, Norwood, MA, USA);Bending test according to PN-EN ISO 7438:2016 [31], which was carried out using the ZD-40 testing machine (AZ Maschinenhandel, Göppingen, Germany);

Fatigue tests were performed using an 8874 INSTRON testing machine using mini samples at room temperature, following the PN-EN 1993-1-9:2007 [32] standard. Hardness testing was performed by PN-EN ISO 9015-1:2011 [33] and PN-EN ISO 6507-1:2024-04 [34] (instead of previous norm: PN-EN ISO 6507-1:2018-05 [35]) standards. The research was supplemented by microstructural results of the weld. First, the resulting segregation were analyzed, and the mechanisms of their nucleation were determined. The analysis of non-metallic inclusions was performed under the microscope of the Zeiss Supra 35 under the EN10247:2017 standard [36]. In addition, a microanalysis of the chemical composition of the precipitates and their impact on the service life of the welded structure was carried out.

The chemical composition of identified inclusions was determined using the EDX detector (Thermo Scientific™, Waltham, MA, USA) with EBSD camera (Orientation Imaging Microscopy v5 Analysis software version 5.31) and OIM Analysis software from EDAX. The results of the study were related to the literature data, and their detailed analysis made it possible to determine the mechanisms of MnS phase formation and complex oxides during the welding of high-strength steels, which has not been described in the literature so far.

## 3. Results and Discussion

### 3.1. Non-Destructive Tests

All joints were subjected to non-destructive testing to check the correctness of the joints and eliminate the joints characterized by inconsistencies and welding defects. The joints made with the X14 and X16 wires were checked by preheating to 115 °C as well as by carrying out the process without preheating. Each sample was appropriately marked (Table 2).

An example of the appearance of a weld made with X16 wire is shown in Figure 3.

The non-destructive tests carried out clearly confirmed that the tested joints did not have cracks or other welding defects in all the analyzed cases. A summary of the results of non-destructive testing (NDT) is presented in Table 3.

At this stage of the research, it can be concluded that the selected parameters and newly developed wires allow the execution of proper welded joints. All joints have been qualified for strength tests.

### 3.2. Destructive Tests

The first research stage was to determine the tensile strength of the welded joint under the PN-EN ISO 527-1 standard. The results of the study are presented in Figure 4.

Analysis of the results allows us to conclude that all joints are characterized by good strength obtained above 750 MPa. Joints made without preheating (Y1 and Y2) have a strength slightly higher than 750 MPa. Figure 4 shows the tensile diagram of the samples, from which the relative elongation can be easily read, too. The maximum relative elongation is registered for the X16 sample. YS was not given, because this boundary is not easy to identify (smooth transition from a straight line to a curve). The ultimate tensile strength (UTS) of the joints is summarized in Table 4.

Strength at this level can be considered sufficient to use such joints on less responsible structures. The Y3 and Y4 samples are characterized by higher strength than the Y1 and Y2 samples. The strength of the weld makes these joints desirable for structural applications in the transportation industry. The strength was above 800 MPa. The information presented in Table 4 also shows that the most durable weld tensile treatment was obtained when both the preheating to 115 °C and laboratory-made electrode wires (with a higher manganese content at the level of 1.6%) were used in the MAG welding process. In the case of unalloyed steels, better strength is related to the lack of preheating (ferritic–pearlitic steels), while heating in HSS steels is aimed at slowing down the rate of cooling of the joint and preventing cracks in the weld. This is related to the structure of HSS steels, which is dominated by the martensitic structure. This is comparable to the results obtained by Górka J. [25] using a welding process with different cooling rates. The authors also indicated that the manganese content in the welding wire translates into obtaining good mechanical properties of the welded joint, which is desired in the automotive industry. Slightly better results were obtained for these connections.

The assumption was confirmed by studies of the weld microstructure, showing the distribution of larger non-metallic precipitates along the grain boundaries and small precipitates inside the dendritic core (Figure 5).

The separations strengthened the structure and translated into its mechanical properties. However, since the literature [19,20,21,22] describes cases of fabrication of structures characterized by non-metallic inclusions of different morphology, shape, and size, which on the one hand improve the tensile strength of the joint and on the other hand reduce its fatigue strength, it was decided to take a broader look at the issue. The results of the study of the chemical composition and morphology of non-metallic inclusions are discussed in detail in the discussion of the research results. The problem is described in terms of the possible mechanisms of formation of the MnS phase separation and the impact of the resulting separation on the operational properties of the joint. Regardless of the results of the structural tests, mechanical tests were carried out, such as bending or fatigue tests of the obtained connections.

The first mechanical tests checked the plastic properties of the joints. The bending tests were carried out from the face and root side. The results of the research are presented in Table 5. After the bend test, the minor cracks from the root side in the Y1 sample were observed. The result is consistent with the literature data, emphasizing the role of preheating in the bonding process of high-strength steels and its impact on the quality of the obtained joints [25,37]. In the remaining tested samples, no cracks were found in both the face and root bending tests.

The results of the study showed that these joints have good plastic properties. No other welding defects were observed after the bending test. The results confirmed that the use of preheating in the welding process of high-strength steel is justified. The next stage of the research was to perform fatigue tests. A Wöhler chart of S700MC welded with two electrode wires was drawn. Fatigue tests were performed for Y3 and Y4 specimens, where two different electrode wires were used during preheating welding (Figure 6). The tests aimed at checking whether the connections had sufficiently high fatigue strength for applications in the automotive industry. We conducted the research for indicative purposes. In our research, we did not focus on the details related to fatigue results, because the results obtained indicated comparable fatigue properties of the joints. We did not note any significant impact of the newly used electrode wires on the fatigue strength of the joints.

A comparison of Wöhler graphs (on a logarithmic scale) shows that the fatigue strength is similar in both cases and amounts to about 420 MPa. A slightly higher fatigue life was determined for the joint made using the X16 electrode wire. Both the results of tensile strength tests (Table 4) and fatigue life indicate slightly better mechanical properties of the joint made with the MAG process using welding wire with a higher manganese content. The tested samples achieved more than two million cycles, which is significantly more than the tested S690QL steel grades presented in [38]. Such conclusions were somewhat expected given the tensile strength of the tested weld. The scattering of the results is most likely due to the high strength of the weld, the microstructure, and the number of ceramic inclusions. For further research, the Y4 sample with the best strength parameters was selected.

For the Y4 joint, the hardness distribution was checked. The results of the tests are presented in the form of a graph of changes in hardness from the center of the weld (Figure 7). The MAG preheated joint using the newly developed X16 welding wire has a slightly lower hardness in the weld seam and heat-affected zone (light blue in the diagram) compared to the base material. We interpret our results as the effect of partial tempering of martensite in the HAZ (Figure 7b). A photo (Figure 7b) is showing the base material (lower left corner), the fusion zone (main diagonal of the rectangle) and the heat-affected zone (upper right corner). In the heat-affected zone, the structure is clearly fragmented, but the martensite may be partially tempered.

The weld has different cooling conditions, and different metallurgical processes take place. It should be emphasized that the results indicate that the recorded difference in the hardness of the joint in its areas is not significant.

The obtained joint is characterized by a comparable level of hardness in the entire cross-section, which, combined with high fatigue strength, is an excellent, desired result compared with the results of the research presented in the articles [27,39].

The result proves the correct selection of welding parameters and the high quality of the joint obtained in the MAG process with preheating and the use of a newly developed welding wire.

### 3.3. Discussion of Research Results—Mechanisms of Inclusions

Analysis of the weld structure shows that numerous non-metallic inclusions occur in the welds under study. The welding wires with increased manganese and silicon content contributed to forming the observed precipitates in the form of oxides, their spinel, or sulfides (Figure 8, Figure 9 and Figure 10).

The mechanisms of formation of oxide inclusions in welds shape the mechanical properties of welded joints. These mechanisms determine the distribution, size, and morphology of oxides. The formed oxides affect the operational properties of the joint. The most observed oxides on a scanning microscope consisted of Mn-Al-Si-O glassy systems. This oxide formation is associated with a high cooling rate of the weld seam (approx. 105 °C/s) during the MAG welding process.

In steel welding processes with the use of wires with a high Mn content, it is impossible to avoid the formation of MnS precipitates. This is mainly due to the strong interaction between Mn-S as well as the difference in energy at the interface of phase separation, matrix metallic material and MnS separation. This energy is much lower than in the case of creating non-metallic oxide inclusions. The literature indicates that MnS can crystallize or precipitate in alloys, depending primarily on the S content [39,40,41,42]. The addition of Mn can reduce the risk of cracking, although these steels are not likely to be exposed to hot cracks. MnS is formed at 1620 °C from liquid L2 (congruently melting point). MnS partially goes into the slag; therefore, it binds sulfur and reduces its concentration in the structure. Manganese is also an effective deoxidizer; it mainly forms MnO at 1790 °C (congruently melting point) and binds oxygen.

With a higher content of S, many more MnS inclusions are formed than with a controlled, reduced S content. However, despite controlling the sulfur content in materials containing manganese, the mechanism of formation of MnS phase precipitations is so strong that in welding processes, it is impossible to avoid the formation of unfavorable precipitates in the weld. Therefore, despite the reduction in sulfur content in the welding wire used in the process, these unfavorable residues were identified in the weld.

It should be emphasized that the identified MnS segregations are characterized by a diverse morphology. The mechanism of morphologically diverse segregation formation has not yet been thoroughly studied and comprehensively discussed. According to the authors, the cause of morphologically different precipitations in welding processes is influenced by interpenetrating mechanisms related to the following [39,40,41,42,43,44,45]:Phase coherence (metallic matrix—MnS; metallic matrix—oxide—MnS);Energy difference at the interface (phases, grains);The percentage of the components concerned (local depletion or excess of the component in cooling speed (heat removal from the weld);The location of the separation (e.g., dendritic core, inter-dendritic boundary).

The results of the research unequivocally confirm that inside the weld in the process of welding with wires with increased manganese content, glassy manganese precipitates (most often inside the grains), patch, shell and mixed MnS inclusions were obtained, and these are presented in Figure 9.

In the zone of the dendritic structure, characterized by elongated grains, mixed precipitates formed on oxides were identified. The dendritic structure is formed because of directional heat outflow from the joint. The resulting elongated grain boundaries are structural errors that disturb energy locally, which promotes the formation of oxide and MnS phases. Microstructure studies have also confirmed that MnS phases precipitate on the surface of previously formed oxides, especially Si oxides (Figure 10).

Such oxides, on the one hand, can strengthen the weld; on the other hand, they can promote the dissolution of Mn and S at elevated temperature and keep these elements even below the freezing point of the alloy, which promotes the nucleation of the MnS phase. It is also exciting that the morphology of MnS on oxides is different, depending on the location of the oxide. The observations confirm the complexity of the processes of non-metallic phase formation in welding processes. They are consistent with the observations made by You et al. [40] and Liu et al. [41]. The authors of the study found that the MnS phase can precipitate by S diffusion from the matrix to the oxide because of segregation during cooling, especially under welding conditions. Therefore, the MnS phase has a much higher tendency to precipitate on oxide during cooling in welding processes. In addition, the cooling processes are slightly changed in different areas of the weld seam. Literature analysis shows that uneven rapid cooling during welding can intensify the diffusion of elements in the material, including sulfur (for example, due to different local chemical compositions at the interdendritic boundary and the dendrite core). Therefore, diffusion will proceed differently around the dendritic structure and within the area characterized by the equiaxial distribution of the grain.

The resulting number, size and morphology of precipitates depend on their location in the weld. In micro-areas, differences in S concentration are recorded. This difference of S concentration creates a local change of environment for the formation of MnS. As the results show, we observed local differences in the microstructure and morphology of MnS. The observed MnS-type oxides are nucleated in dendrite-type grains. The micro-area shows depleted concentrations of S. On the other hand, complex MnS precipitates (simultaneous combination of the coating and patch types), such as those shown in Figure 8, could only have formed in the case of earlier formulation of the oxide (oxide spinel), leading to drastic local changes in the chemical composition. Intensified diffusion processes in these regions were the primary mechanism influencing the formation of segregations with complex morphology.

### 3.4. Morphology of MnS Weld Inclusions and Exploitation Durability of Structure

MnS inclusions have a significant impact on the welded structure service life [39,40,41,42,43]. If welded structures are deformed at a low deformation rate, micro-cracks may form, and the displacement of the crack apex opening will be affected by the low interfacial energy between the metallic matrix and the inclusion, thus weakening the entire structure. On the other hand, the decohesion between the matrix and inclusions, e.g., complex MnS evolved on an oxide inclusion, can have a negative effect in the case of significant deformation velocities, which can be observed in the results of impact tests.

Therefore, both the type of inclusion and its morphology of forming, e.g., the formation of MnS on the oxide, can be an essential factor influencing the mechanical properties of the welded joint. The studies confirmed that the different morphology of MnS is formed in welding conditions. According to literature data [43], this phase hurts the properties of welded joints. Therefore, it justified reducing the sulfur content in the joint, which is aimed at reducing the amount of unfavorable MnS inclusions.

Most of the non-metallic inclusions identified in the weld were oxides or their spinels and small residues of the MnS phase. These unfavorable inclusions were created in rapidly changing thermodynamic conditions related to the directional heat dissipation from the weld in the welding process. They are consistent with the observations made by Li et al. [44] and Wu et al. [45]. The amount and distribution of these oxides allow us to conclude that they do not weaken the structure. Identified inclusions do not pose a threat to the safety of the welded structure.

The structure of the weld is characterized by a small number of complex inclusions concerning oxide and patch MnS precipitates. The MnS releases formed in the analyzed welding process, due to their relatively small number and distribution (lack of clusters), should not significantly affect the service life of the obtained joints. In the analyzed case, this was also confirmed by the results of strength tests.

## 4. Conclusions

This paper presents a new concept for welding HSS steels. The aim of this manuscript was to check whether the newly developed electrode wires enable correct welded joints to be made and whether their chemical composition will affect the quality and strength of the obtained joints. It was proposed to use electrode wires (laboratory-made) with a high content of silicon (0.9%) and manganese (1.4% Mn and 1.6% Mn) to obtain good mechanical properties for the joint. The S700MC welding process is designed for applications in the automotive industry.

The results confirm good properties and quality for the obtained joints:NDT tests showed the absence of welding defects and inconsistencies in all cases analyzed.Tensile strength tests have shown that preheating before welding S700MC steel is beneficial and increases the UTS value from 750 to approx. 820 MPa.Similarly, the bending test reinforces the belief that junctions have better plastic properties after preheating.On the other hand, the fatigue tests showed that all tested joints are at a similar level made with two welding wires with different manganese content. In all cases, a high fatigue strength value of 420 MPa was obtained.

It has been shown that a higher manganese content of 1.6% and the use of a preheating temperature of 115 °C allow for obtaining a high tensile strength of the joint at the level of 820 MPa and, at the same time, a high fatigue value at the level of about 420 MPa, which is at a higher level than that obtained by authors of other studies on the weldability of S700 MC steel. Summarizing the test results, it can be noted that both high tensile strength and good fatigue strength make joints made with newly developed electrode wires suitable for use in the automotive industry.

The strength of the joints is determined by the obtained structure, which is characterized by the occurrence of evenly distributed non-metallic oxide inclusions. Additionally, the mechanism of inclusion formation during welding processes using newly developed electrode wires was determined.

The identified oxides strengthen the structure, translating into significant joint strength above 800 MPa. In addition, morphologically different inclusions of the MnS phase were identified in the structure:It was found that the morphology of the precipitates and their coherence with the matrix can affect both the weakening and strengthening of the operational strength of the weld depending on the deformation forces acting on the structure in use. A different type of precipitation in the weld is desirable for a structure subject to long-term deformations, and another kind is desirable for fast-deformable structures.The operational life of welded joints is also determined by the number of inclusions in the weld, separations between the oxides and sulfides and their mutual location. Local sediment concentration is not desirable. This concentration creates the possible cracking of the structure during use. The morphology of the inclusions and their coherence with the matrix can, therefore, both weaken and strengthen the operational strength of the weld.

It can be concluded that the proper selection of welding parameters should translate into obtaining a structure characterized by low MnS content and evenly distributed oxides in the weld, which translates into an increase in the level of strength of the welded joint in strength tests.

Analysis of the test results allows us to conclude the following:For obtaining a welded structure made of S700 MC steel, with high strength, preheating to 115 °C and an electrode wire with increased manganese and silicon content should be used;The hardening effect in the weld structure is observed when silicon and aluminum oxides and oxide spinels are evenly distributed and not too large on oxides containing silicon in dendritic areas; then, unfavorable MnS phases may be formed, weakening the weld, as these oxides promote the dissolution of Mn and S at high temperatures of the welding process;Rapid heat dissipation from the weld promotes the formation of amorphous, glassy precipitates.

## Figures and Tables

**Figure 1 materials-17-02974-f001:**
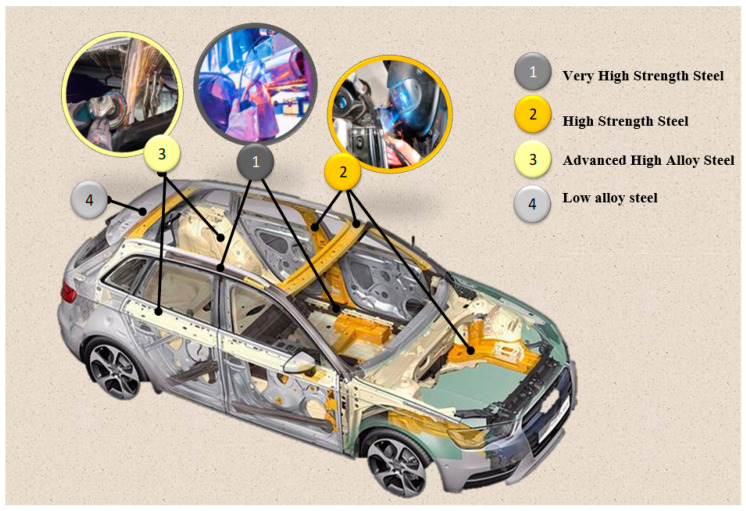
The use of several types of steel in the welding construction of vehicle bodies (own study based on [11]).

**Figure 2 materials-17-02974-f002:**
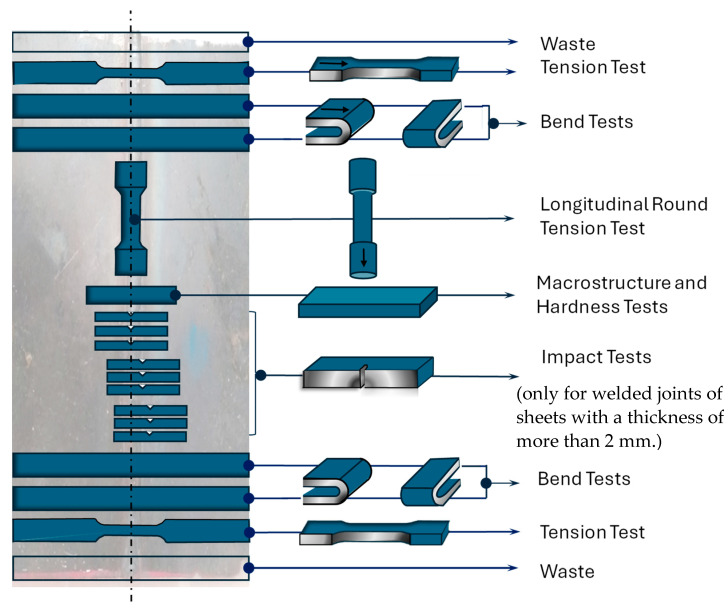
Scheme of sampling for testing (own research).

**Figure 3 materials-17-02974-f003:**
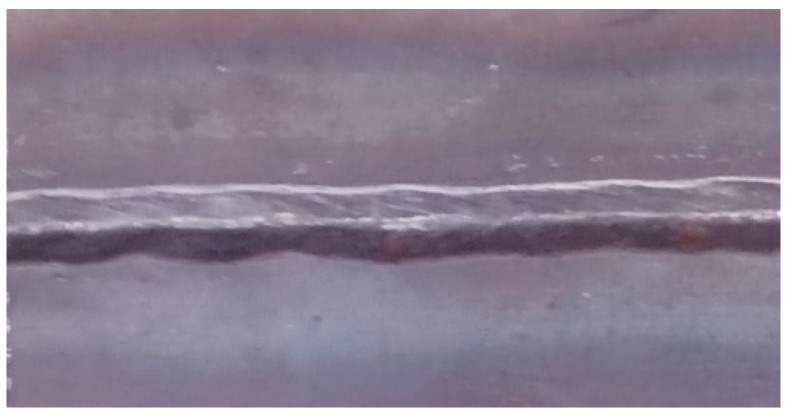
Single-pass weld joint obtained by MAG (Metal Active Gas) process with X16 wire.

**Figure 4 materials-17-02974-f004:**
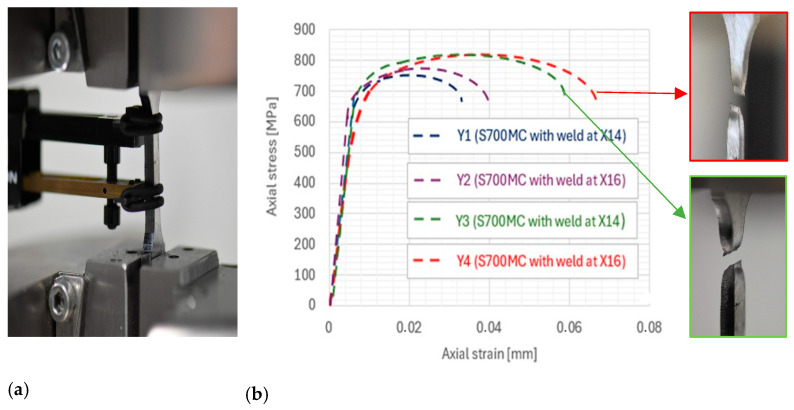
Single-pass weld joint obtained by MAG (Metal Active Gas) process with X16 and X14 wires: (**a**) the sample in the machine—view, (**b**) strength results diagram and view of selected samples after test.

**Figure 5 materials-17-02974-f005:**
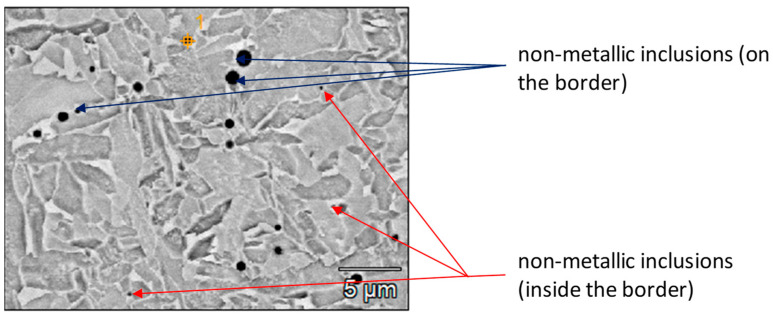
Non-metallic inclusion in the weld structure.

**Figure 6 materials-17-02974-f006:**
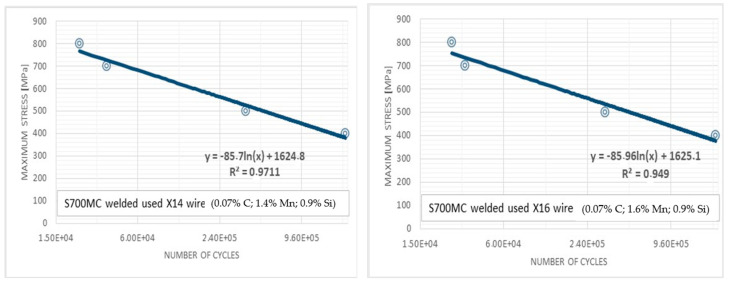
Results of fatigue tests obtained for joints made with the X14 (**left**) and X16 (**right**) electrode wires.

**Figure 7 materials-17-02974-f007:**
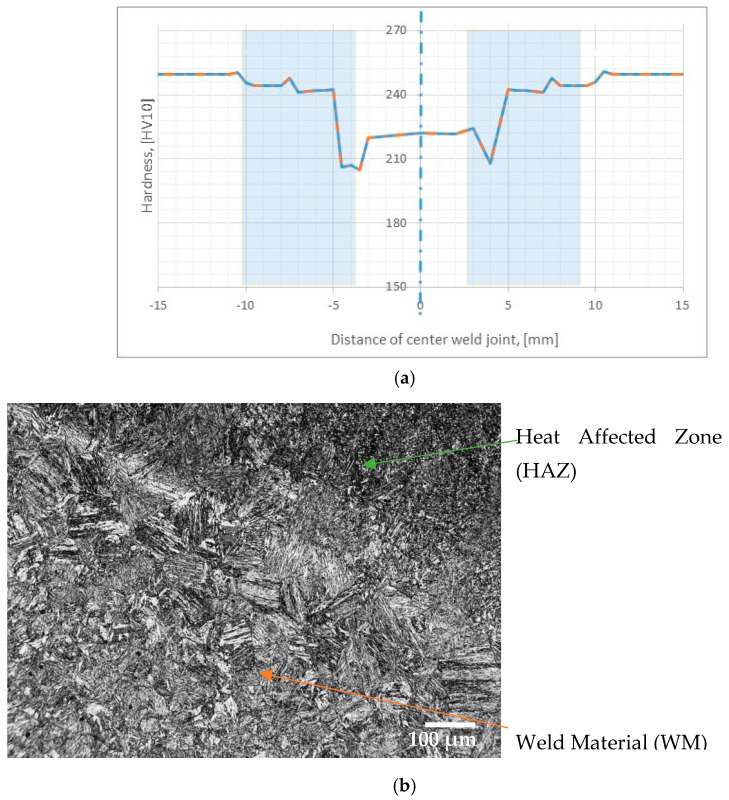
Y4 joint (**a**) hardness measurements, (**b**) microstructure.

**Figure 8 materials-17-02974-f008:**
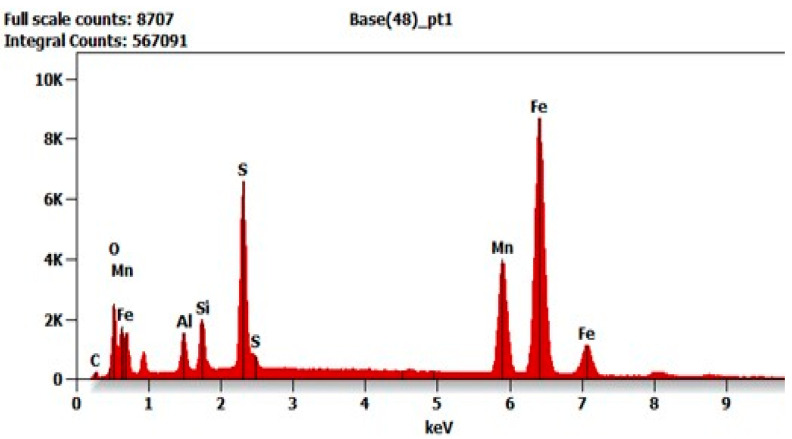
X-ray microanalysis of the non-metallic inclusion at point (1) in Figure 5.

**Figure 9 materials-17-02974-f009:**
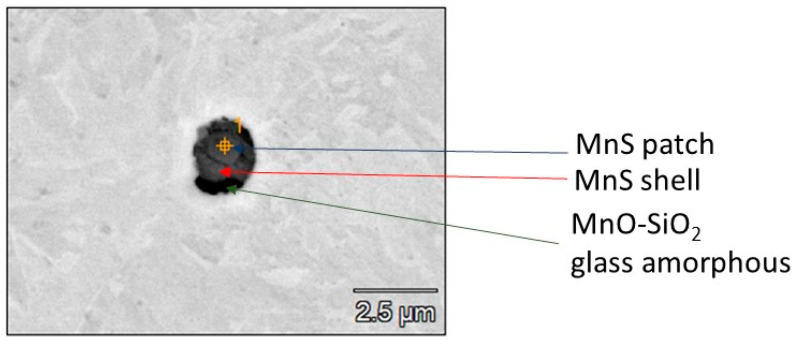
The morphology of MnS inclusion. It was found in the dendritic area structure.

**Figure 10 materials-17-02974-f010:**
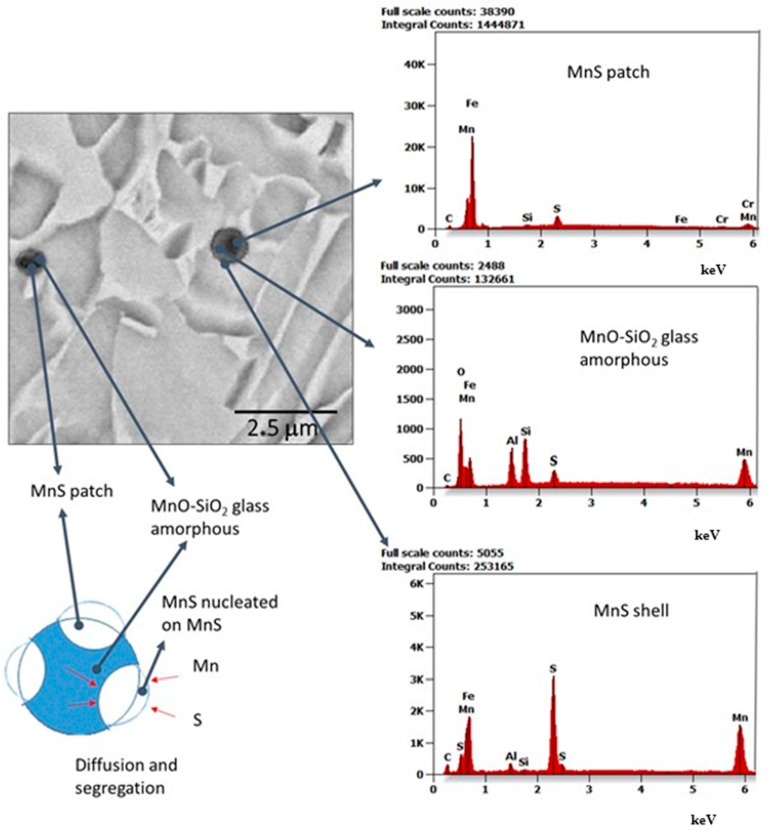
Heterogenous nature of non-metallic inclusion.

**Table 1 materials-17-02974-t001:** Chemical composition and tensile strength of the S700MC steel: YS—yield stress, UTS—ultimate tensile strength.

Chemical Composition of the S700MC Steel (%)								Tensile Strength (MPa)	
C	Si	Mn	P	S	Al	Nb	Ti	YS	UTS
0.12	0.92	2.2	0.023	0.011	0.01	0.1	0.15	690	970

**Table 2 materials-17-02974-t002:** Specimen designations.

Specimen	Welding Wire	Preheating Temperature, °C
Y1	X14	20
Y2	X16	20
Y3	X14	115
Y4	X16	115

**Table 3 materials-17-02974-t003:** Specimen designation.

Specimen	Inspection Results:	
Y1	No defects, no cracks	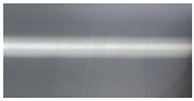
Y2	No defects, no cracks	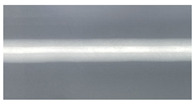
Y3	No defects, no cracks	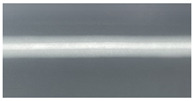
Y4	No defects, no cracks	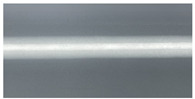

**Table 4 materials-17-02974-t004:** Ultimate tensile strength (UTS) of the joints (average of 3 measurements).

Specimen	UTS, MPa
Y1	753
Y2	767
Y3	817
Y4	823

**Table 5 materials-17-02974-t005:** Results of a bending test of the weld.

Specimen	Face Side		Root Side	
Y1	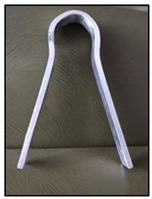	No cracks	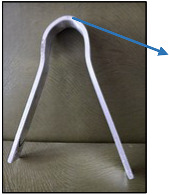	slight cracks 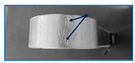
Y2	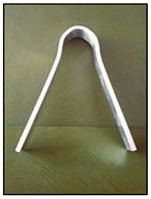	No cracks	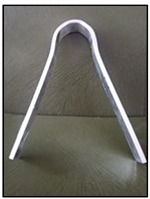	No cracks
Y3	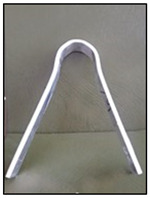	No cracks	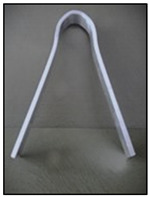	No cracks
Y4	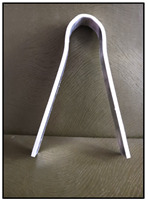	No cracks	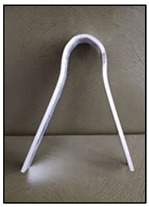	No cracks

## Data Availability

The original contributions presented in the study are included in the article, further inquiries can be directed to the corresponding authors.

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
