# Peer review of "Formation of Oxides and Sulfides during the Welding Process of S700MC Steel by Using New Electrodes Wires"

_materials, 2024, doi:10.3390/ma17122974_

Round 1
Reviewer 1 Report
Comments and Suggestions for Authors
General comment: The manuscript seems like a report. The results was only presented without a clear and deep discussion.
1. The manuscript title did not summarize the entire work. The title must be rewritten.
2. The introduction section did not provide a clear justification of the why the manuscript is important.
3. The manuscript objective is not clear.
4. Perform a short review (~ 1 page) about the effect of alloying elements, in special the Si, on the microstructure and mechanical properties the weld metal. In addition, what aspects were considered to use a Si-rich filler metal? This is the most important point of the manuscript.
5. Improve images quality.
6. Add on Table 4 the ductility and yield strength.
7. The authors must explain why a higher “welding temperature” indices a higher UTS. In general, a slow cooling rate (higher welding temperature) induces a soft microstructure.
8. “The assumption was confirmed by studies of the weld microstructure, showing the distribution of larger non-metallic precipitates along the grain boundaries and small precipitates inside the dendritic core (Figure 5).” For a low-alloy steel, it is not possible to observe the dendritic core. This is a transformable alloy, so three phase transformation occurred from solidification to room temperature, e.g., liquid → δ, δ → γ, γ → α. The non-metallic precipitates formation and its position must better described.
9. The fatigue data is superficially described (both the experimental procedure and data description and discussion). Add the statistical data of the Fatigue results. In addition, from Wöhler graphs, used the Morrow’s equation to better describe the fatigue behavior of the studied alloys.
10. Better describe the hardness data. In figure 7, add all hardness profile (4 welded joints). In addition, in this figure, indicate the zones of welded joint (base metal, HAZ, and weld metal).
11. Despite the authors pointed out that the main problem related with the welding of martensite low-alloy steel was the low strength of the weld metal. Typically, it is observed the softening of the HAZ, as also verified in Figure 7. So, how to overcome the HAZ softening problem?
12. The authors must add some micrographs about the weld metal and HAZ microstructure. And discuss the effect of each microstructure in the final mechanical behavior.
13. The conclusion must be focused and concise. In addition, as the authors did not define the manuscript objective, it is impossible to conclude.
Author Response
Thank you very much for you kind review and suggestions. We gave the answers in the atachment.

Reviewer 2 Report
Comments and Suggestions for Authors
The authors presented very good research on S700MC steel and its most important properties when it is used in welded structures. The selected group of steels has increasing significance in the industry. The paper is well-conceived, the experimental plan is complete, and the results are presented appropriately.
My comments are listed below:
-
In Section 2, the electrode designations X16 and X14 are given, but the standard by which these designations are defined is not mentioned. The designation according to an international standard should be added.
-
How was the preheating temperature determined? Manufacturer recommendations, author experience, or calculation (as in the paper http://dx.doi.org/10.1016/j.matdes.2009.11.066 or in https://doi.org/10.3390/ma15093082)?
-
For the shielding gas flow rate in welding, the unit l/min is more commonly used instead of dm³/min.
-
The font size for the unit of shielding gas flow and then the font size of some words in the next sentence is larger than the base font size.
-
In Section 2, it is not mentioned whether the welding was performed manually or by a robot.
-
In Section 3 - Results, below Figure 4, a photograph of the samples after the tensile test should be presented (similar to what was done for the bending test) to show the zone of the joint where the fracture occurred.
-
Below Figure 6, a paragraph analyzing the fatigue properties of the base material should be added, e.g., as done in the paper https://doi.org/10.3390/met12071199 for S690QL steel with similar characteristics. Perhaps you can refer to this study?
-
The paper should undergo another round of technical editing as there are errors such as the missing period after the word "drowned" in line 235 or an extra space in the title 3.3 at the end of the last word.
-
I suggest moving Figure 8 to page 9 immediately after the first paragraph in section 3.3 because it is mentioned there for the first time.
-
In section 3.3, the harmful effect of sulfur-based inclusions is mentioned, but hot cracks that excess sulfur can cause are not mentioned. Is this steel prone to hot cracking or not? I believe that the addition of Mn reduces the risk of hot cracking.
Once more I want to emphasize that I find the research very good and important for practical aspects.
Comments on the Quality of English Language
The authors presented very good research on S700MC steel and its most important properties when it is used in welded structures. The selected group of steels has increasing significance in the industry. The paper is well-conceived, the experimental plan is complete, and the results are presented appropriately.
My comments are listed below:
-
In Section 2, the electrode designations X16 and X14 are given, but the standard by which these designations are defined is not mentioned. The designation according to an international standard should be added.
-
How was the preheating temperature determined? Manufacturer recommendations, author experience, or calculation (as in the paper http://dx.doi.org/10.1016/j.matdes.2009.11.066 or in https://doi.org/10.3390/ma15093082)?
-
For the shielding gas flow rate in welding, the unit l/min is more commonly used instead of dm³/min.
-
The font size for the unit of shielding gas flow and then the font size of some words in the next sentence is larger than the base font size.
-
In Section 2, it is not mentioned whether the welding was performed manually or by a robot.
-
In Section 3 - Results, below Figure 4, a photograph of the samples after the tensile test should be presented (similar to what was done for the bending test) to show the zone of the joint where the fracture occurred.
-
Below Figure 6, a paragraph analyzing the fatigue properties of the base material should be added, e.g., as done in the paper https://doi.org/10.3390/met12071199 for S690QL steel with similar characteristics. Perhaps you can refer to this study?
-
The paper should undergo another round of technical editing as there are errors such as the missing period after the word "drowned" in line 235 or an extra space in the title 3.3 at the end of the last word.
-
I suggest moving Figure 8 to page 9 immediately after the first paragraph in section 3.3 because it is mentioned there for the first time.
-
In section 3.3, the harmful effect of sulfur-based inclusions is mentioned, but hot cracks that excess sulfur can cause are not mentioned. Is this steel prone to hot cracking or not? I believe that the addition of Mn reduces the risk of hot cracking.
Once more I want to emphasize that I find the research very good and important for practical aspects.
Author Response

(The authors gave the same response as above.)

Reviewer 3 Report
Comments and Suggestions for Authors
Dear Authors,
Please find my comments in the attached pdf.

Author Response

(The authors gave the same response as above.)

Reviewer 4 Report
Comments and Suggestions for Authors
In this study, newly developed electrode wires with increased silicon and manganese content were used to receive a high-quality welding structure of high-strength S700MC steel for applications in the automotive industry. Overall, it is a good work. Minor revision is required before publication:
(1) Fig4 and Tab4, it seems the tensile strength test was only conducted once for each. It is suggested to test at least 3 times.
(2) Caption of Fig. 8, should be at point (1) in Figure 5? Please check.
(3) English and format should be checked carefully.
Author Response

(The authors gave the same response as above.)

Round 2
Reviewer 1 Report
Comments and Suggestions for Authors
The authors improve the manuscript quality
Reviewer 3 Report
Comments and Suggestions for Authors
Much better, accepted.